# Fabrication of Silver Nanobowl Arrays on Patterned Sapphire Substrate for Surface-Enhanced Raman Scattering

**DOI:** 10.3390/mi14061197

**Published:** 2023-06-05

**Authors:** Yanzhao Pang, Mingliang Jin

**Affiliations:** 1South China Academy of Advanced Optoelectronics, South China Normal University, Guangzhou 510006, China; pangyz@m.scnu.edu.cn; 2International Academy of Optoelectronics at Zhaoqing, South China Normal University, Zhaoqing 526060, China

**Keywords:** SERS, Ag nanobowl, self-assembly, PSS, displacement reaction

## Abstract

The current article discusses surface-enhanced Raman spectroscopy (SERS) as a powerful technique for detecting molecules or ions by analyzing their molecular vibration signals for fingerprint peak recognition. We utilized a patterned sapphire substrate (PSS) featuring periodic micron cone arrays. Subsequently, we prepared a three-dimensional (3D) PSS-loaded regular Ag nanobowls (AgNBs) array using self-assembly and surface galvanic displacement reactions based on polystyrene (PS) nanospheres. The SERS performance and structure of the nanobowl arrays were optimized by manipulating the reaction time. We discovered that the PSS substrates featuring periodic patterns exhibited superior light-trapping effects compared to the planar substrates. The SERS performance of the prepared AgNBs-PSS substrates was tested under the optimized experimental parameters with 4-mercaptobenzoic acid (4-MBA) as the probe molecule, and the enhancement factor (EF) was calculated to be 8.96 × 10^4^. Finite-difference time-domain (FDTD) simulations were conducted to explain that the AgNBs arrays’ hot spots were distributed at the bowl wall locations. Overall, the current research offers a potential route for developing high-performance, low-cost 3D SERS substrates.

## 1. Introduction

Surface-enhanced Raman spectroscopy (SERS) is a powerful analytical technique that detects molecules or ions by analyzing their molecular vibration signals for fingerprint peak recognition [1]. This approach enables rapid, highly sensitive, label-free, and multi-channel detection of probe molecules through amplifying Raman scattering signals in the “hot spot” region of SERS enhancement, compared to regular Raman detection [2,3]. According to the electromagnetic enhancement theory, the interaction between incident light and a metal surface induces a collective resonance of free electrons, generating plasmonic excitations [4]. These excitations give rise to local surface plasmon resonance (LSPR) within the nanostructure regime [5]. Specifically, areas characterized by nano-gaps, nano-slits, and nano-tips with dimensions smaller than 10 nm are identified as “hot spots” exhibiting the most pronounced LSPR effects. LSPR significantly amplifies the electric field in the vicinity of the metal surface. A molecule as a point dipole that responds to the enhanced local electric fields at or near the metal surface enormously boosts the Raman signal [6]. As such, SERS spectroscopy has emerged as a valuable analytical tool widely used in various fields, including biomedical research [7], analytical science [8,9], and material physics [10], and has garnered growing research interest. The acquisition of SERS spectra is significantly influenced by the performance of SERS substrates, which generally fall into two main categories: colloidal nanoparticles and chip-based substrates, including two-dimensional (2D) or three-dimensional (3D) nanoarrays. Colloidal metal nanoparticles’ electromagnetic field enhancement factor (EF) can sometimes reach 10^12^. Yet, their structure and SERS signal stability are often suboptimal because they cannot trap the target molecule firmly in the “hot spot” [11]. In contrast, chip-based substrates can be engineered with reliable periodic nanostructures that provide uniform Raman signals. With significant advancements in scientific principles and manufacturing technologies, SERS substrates have undergone an evolution from 2D nanoarrays to 3D structures [1].

The utilization of a confocal laser beam in the SERS spectrometer reveals that 2D substrates fail to maximize the excitation light due to the 3D cone shape of the beam with a height of several tens of microns [12]. In contrast, when interacting with the same irradiated volume, the three-dimensional structure of the substrate substantially amplifies the surface area and the number of spatially distributed hot spots that engage with the laser. This enhancement in surface area and “hot spot” density increases interactions and provides more potential for intricate laser–substrate interactions within the irradiated volume. These 3D substrates offer additional side surfaces and a larger total surface area, capturing more target molecules in the hot spot region. Furthermore, 3D nanostructures in vertical coupling exhibit new plasmon modes, achieving higher hot spot density and better laser utilization efficiency with the same confocal laser beam [13]. Three-dimensional (3D) SERS substrates possess the remarkable advantage of customizable shape, size, and structure periodicity along the *z*-axis. This flexibility enables the manipulation of optical phenomena and plasmonic properties by increasing the structural complexity. For instance, incorporating a vertical stacking plasmonic structure within 3D multilayer substrates offers enhanced performance. Similarly, 3D microporous substrates provide a substantial internal surface area, while the ability to implement controllable and ordered structures in 3D array substrates further adds to their versatility. Hence, these substrates offer ample opportunities for excellent SERS detection. Currently, many researchers pay more attention to the “hot spot” structure of the metal layer on the SERS substrate, ignoring the light-harvesting efficiency of the supporting substrate. Moreover, it has been suggested that a light scatterer deflects the incident light to increase the optical path when using a patterned sapphire substrate (PSS) coated with metal, providing excellent optical capture capability and higher Raman enhancement [14,15,16]. The patterned sapphire substrate (PSS) is identified as a potential carrier for preparing low-cost, high-performance 3D SERS substrates due to their commercial viability and low cost [17,18,19].

Unlike nanoparticle structures, noble-metal nanobowl (NB) structures can more effectively bind multiple forms of the “hot spot” to enhance their surface-localized electric-field strength and produce powerful Raman signals [20,21,22,23,24,25]. Simulation calculations indicate that the NB’s upper edge and bottom position can cause lightning-rod effects and resonance-coupling effects to produce large enhancement fields [26,27]. Furthermore, the functionalized nanobowl structures can effectively capture target molecules, and their combination with 3D NB designs can improve the sensitivity of SERS detection [28]. Researchers employed self-assembly and atomic-layer deposition techniques to prepare highly ordered Ag particles in Au/Al_2_O_3_ NB Raman detection sensitivity and biosensing applications [29,30].

To further enhance the plasmon-resonance effect of AgNBs arrays by exploiting the light-harvesting effect of 3D substrates, this paper presents a method for preparing a 3D SERS substrate with PSS-supported regular AgNBs structures using polystyrene (PS) nanospheres, self-assembly, and surface electrochemical displacement reactions. The PSS is loaded with high-density AgNBs structures after removing the PS spheres from the Ag@PS core-shell composite nanospheres. The reaction time parameters are optimized to improve the hot spot structure and SERS performance. The optimized experimental parameters detect 4-mercaptobenzoic acid (4-MBA) as the probe molecule with a Raman signal increased by almost 10^5^ times.

## 2. Materials and Methods

### 2.1. Preparation of PSS-Loaded Ag Nanobowl Structures

Preparation of the PS nanoparticle solution: A centrifuge tube was loaded with a 10 wt.% PS nanoparticle aqueous dispersion of 150 nm in size. After ultra-sonic cleaning and centrifugation, an equal volume of ethanol and water were added to the dispersion to yield a solution with a specific mass fraction of C_2_H_5_OH:H_2_O and a volume ratio of 1.

Self-assembly of PS spheres by the drainage method: A freshly cleaned silicon wafer measuring 2 cm × 2 cm was positioned at the bottom of an inner pool. The PS nanoparticle, ethanol, and water dispersion was introduced onto the water surface with an injection pump at 2 μL/min. Following drainage and drying, a self-assembled monolayer film of PS nanoparticles was generated on the silicon wafer using the drainage method.

Adsorption of Sn^2+^ on the PS nanoparticle film and transfer to the silicon wafer: A SnCl_2_ solution with a volume of 150 mL and a concentration of 13.3 mM was added to the water pool following drainage. The silicon wafer with the self-assembled monolayer film of PS nanoparticles was placed at a 45° angle onto the water surface, leading to the detachment and re-entry of the PS monolayer film onto the water surface, which facilitated the adsorption of Sn^2+^ from the water solution. After 5 min, the water pool was activated, and following drainage and drying, a self-assembled monolayer film of PS nanoparticles with adsorbed Sn^2+^ was formed on the silicon wafer.

The substitution–reduction reaction of Ag on the surface of PS nanoparticles: An NH_4_OH solution of 0.12 mM concentration, an AgNO_3_ solution of 0.06 mM concentration, and a C_6_H_5_Na_3_O_7_·2H_2_O (TSC) solution of 0.06 mM concentration were introduced to the reaction pool. The PSS with the microcones facing upwards was pre-placed on the bottom of the reaction pool. The silicon wafer with the self-assembled monolayer film of PS nanoparticles with adsorbed Sn^2+^ was positioned at a 45° angle onto the liquid surface. This action resulted in the detachment and re-entry of the PS monolayer film onto the water surface, allowing Sn^2+^ on the surface of the PS nanoparticles to react with Ag[(NH_3_)_2_]^+^ in the water solution. Due to the buoyancy exhibited by the PS nanospheres on the liquid surface, the silver shell formation is observed exclusively on the lower hemisphere, which comes into contact with the silver–ammonia reaction solution. By promoting this reaction, TSC facilitates the transfer of silver ions onto the PS surface, forming a stable silver layer. The flow rate of the reaction solution was regulated to control the reaction time, which enabled the formation of an Ag film layer on the surface of the PS nanoparticles. After drying for about 15 min, a self-assembled monolayer film of PS nanoparticles with a surface Ag layer was formed on the PSS. The PSS with periodic microcones distributed on the upper surface was purchased from Guangdong Zhongtu Semiconductor Technology Co., Ltd. (Dongguan, China). The height of the top of the microcone is 1.8 μm, the diameter of the bottom of the microcone is 2.8 μm, and the distance between the centers of the microcones is 3 μm.

Removal of surface PS nanoparticles: The PSS with the self-assembled monolayer film of PS nanoparticles possessing a surface Ag layer was immersed in a DMF solution at 60 °C for 8 min to dissolve the PS nanoparticles. This process yielded an array of densely packed Ag nanobowls (AgNBs) on the microtip of the PSS, which was named AgNBs-PSS. The sample was rinsed with water, dried with N_2_, and stored in a vacuum. It was then sectioned into 5 mm × 5 mm pieces for subsequent testing. The preparation process is depicted in Figure 1.

### 2.2. SERS Testing and Structural Characterization of the Substrate

Raman measurement of Rhodamine 6G was done by the following process. An amount of 25 μL of a solution containing 10^−7^ M probe molecules was dropped onto the AgNBs-PSS substrate, dried in the air, and washed with deionized (DI) water three times. SERS testing was carried out with a dry substrate. A Finder Insight Raman spectrometer (FI532W) was used for the SERS measurement, with a specific excitation wavelength of 532 nm. The laser beam was focused on the sample through a 50× (NA = 0.55) objective lens, with a focused spot diameter of 10 μm. The laser power was 0.4 mW, and the Raman integration time was 0.3 s. Three positions were selected for each sample, and the SERS was tested three times.

Raman measurements of 4-MBA were done by the following process. The prepared AgNBs-PSS substrate was immersed in the 0.1 M 4-MBA ethanol solution for about 10 h to form a highly packed monolayer of probe molecules. Then the substrate was rinsed with ethanol (99%) 3 times and blow-dried with N_2_ gas. SERS testing was carried out with a dry substrate. The 1 g bulk 4-MBA sample was first solubilized in 1.5 mL ethanol (99%). A 10 μL solution was cast on the Si wafer to form a solid pack of 4-MBA for the reference Raman measurement. The laser power was 6 mW, and the Raman integration time was 1 s.

The surface morphology and local-nanostructure size analysis of the substrate prepared on the silicon wafer were performed using scanning electron microscopy (SEM).

### 2.3. Time-Domain Finite-Difference Simulation

Finite-difference time-Domain (FDTD) analysis simulations were performed using the Ansys Lumerical Finite-Difference IDE software. The model of the AgNBs structure is shown in Figure 2. The software flow is as follows: the FDTD simulation area is a 3D system, and the background dielectric constant is set to air. The global mesh size is set to 5 nm, and an overlay mesh size of 0.3 nm is added near hot spot areas. The boundary condition is a perfectly matched layer with complete absorption. The excitation source was chosen as a global scattering source (532 nm), with the incident light propagating in the negative *z*-axis direction and the polarized light along the *x*-direction. Using a field monitor, the surrounding enhanced electric field intensity is recorded and extracted at the structure’s XY and YZ section locations.

## 3. Results and Discussion

### 3.1. Structure and Optical Properties of PSS

The present study reports on a periodically structured substrate’s characterization and optical properties featuring micro-sized conical bodies. The PSS support substrate, depicted in Figure 3b, exhibits a periodic arrangement of conical structures characterized by a bottom diameter of 2.7 μm, a cone height of 1.7 μm, a period of 3 μm, and a minimum gap of 0.3 μm between adjacent conical bases. The conical bodies have a smooth surface, with a circular bottom surface area of 1.83π μm^2^ and a lateral surface area of 3.04π μm^2^. To assess the performance of the PSS substrate, the reflectance spectra of three samples based on a flat sapphire substrate (SS)–Al mirror substrates, including PSS, a Ag@PS-PSS monolayer film substrate, and a AgNBs-PSS substrate loaded with Ag nano cups, were measured using Ocean Optics USB2000+ spectrometers and microscopes, as presented in Figure 3a. It can be seen that the absorbance of the PSS with a periodic micron cone structure is significantly increased compared to the SS substrate with a planar configuration, and the reflectance is further increased after loading the AgNBs structure. In addition, two absorption peaks are generated due to the presence of the PSS structure at 475 nm and 620 nm. At 620 nm, the positions of the absorption peaks of the PSS substrate and the PS@Ag-PSS substrate are the same, and the slight redshift is due to the metallic nanostructure of the Ag-encapsulated PS spheres on the PSS surface, while the significant decrease in reflectance is due to the nanostructure further enhancing the light-trapping effect. In comparison, the reflectance at 475 nm decreases significantly, which we guess is due to the formation of surface-plasmonic excitations caused by the cavity of the nanobowl. Although the laser wavelength of 532 nm during SERS detection is not at the two lowest reflectivity positions, the absorption in its band is still close to 90% of the trapped light effect. This structure is very favorable to the absorption of SERS excitation light, increasing the electric field strength (E) of the LSPR. According to the electromagnetic enhancement theory of SERS, the enhancement factor (EF) is proportional to the fourth power of the electric field |E/E_0_|^4^ of the localized surface, highlighting that the enhancement of the polarization effect is one of the ways to augment the SERS detection ability.

### 3.2. Preparation of AgNBs Structure

Figure 1 depicts the preparation of AgNBs immobilized on a PSS substrate by a process involving the self-assembly of PS colloidal spheres and surface electrochemical displacement reactions. Table 1 shows the characteristics of the SERS substrates prepared for our work. PS colloidal spheres, which have carboxyl groups on the surface and exhibit a negative charge in an aqueous solution, are self-assembled into a monolayer film on the water surface by floating on an ethanol dispersion solution. This self-assembly occurs due to the minimization of Gibbs’s free energy, resulting in a uniform and stable structure. Electrostatic adsorption between the Sn^2+^ and PS colloidal spheres sensitizes the surface of the spheres and causes them to carry Sn^2+^. The electrochemical displacement reaction between Sn^2+^ ions and [Ag(NH_3_)_2_]^+^ reduces Ag^+^ in the silver–ammonia solution, resulting in the uniform deposition of Ag nanoparticles on the surface of the PS colloidal spheres. The resulting composite nanostructures of Ag-coated PS spheres have been characterized, as shown in Figure 4c,e. The electrochemical displacement reaction is represented by the reaction equation [31].
(1)$${\mathrm{Sn}}^{2+}+2{\left[\mathrm{Ag}{\left({\mathrm{NH}}_{3}\right)}_{2}\right]}^{+}+4H_{2}O\to {\mathrm{Sn}}^{4+}+2\mathrm{Ag}↓+4{\mathrm{NH}}_{3}\cdot H_{2}O$$

### 3.3. Effect of Reaction Time on the Structure and SERS Performance of the Substrate

Under the same reaction conditions, after reacting for 45 min, 1 h, 2 h, 3 h, and 4 h, dry self-assembled monolayers of PS nanospheres with surface Ag layers were formed on the PSS substrate. After removing the PS, the PSS substrate with AgNBs structures was obtained. The substrate was rinsed with pure water, dried with N_2_, and stored in a vacuum. The structure of the resulting nanobowl array is shown schematically in Figure 2. The morphology of the substrate is shown in Figure 5. The structure of the NBs is strongly restricted by the reaction time, with longer reaction times resulting in thicker wall thickness.

Table 2 presents the calculated values of the inner-pore diameter (d), outer diameter (D), and the distance between adjacent nanobowls (P) of AgNBs formed using 150 nm PS spheres at different reaction times. The results indicate that the outer diameter of AgNBs remains relatively constant as the reaction time increases. However, the inner diameter gradually decreases over time. It is important to note that the change in inner diameter is not solely caused by the gradual decrease in the diameter of the circle perpendicular to the spherical center distance. This is evident as the calculated outer diameter (D_w_), corresponding to the height cross-section, is significantly larger than the measured inner diameter (d). The primary factor contributing to the gradual decrease in the inner diameter of AgNBs with reaction time is the electrically coupled replacement reaction of [Ag(NH_3_)_2_]^+^ with Sn^2+^. This reaction involves gradually replacing and reducing Sn^2+^ ions adsorbed on the PS surface to Ag atoms, occurring from the outer region to the inner region. Consequently, a shorter reaction time facilitates the formation of thin-walled Ag nanoshells with greater ease.

SERS tests were conducted using the R6G probe molecule on substrates prepared with different reaction times. SERS spectra are shown in Figure 6. The 45 min, 1 h, and 4 h substrates exhibited strong Raman signals, with peak intensities of 550 counts, 486 counts, and 479 counts at 612 cm^−1^, respectively. The 2 h reaction substrate had the lowest intensity at 612 cm^−1^, likely due to the increase in shell thickness and the decrease in bowl wall height. Increasing the reaction time beyond 2 h transformed the nanobowl structure into a nanocolumn structure, which increased the number of hot spots on the outer wall. It can be seen that with the further increase of the reaction time to 3 h or 4 h, the bowl wall height gradually increases and transforms the lamellar nanopore structure. The nanopore structure also makes it easier for the form of enrichment of the substance to be detected, thus increasing the number of enhanced hot spots in the pore. These results provide valuable insights into the effects of electrodeposition reaction time on the size and morphology of AgNBs-PSS substrates and their potential use in SERS applications. The elevated nanobowl coverage effectively enhances the density of hot spots, thereby contributing to the enhanced reliability of single SERS signal detection and the substrate’s signal repeatability. However, it is important to note that the hot spot density has a constrained impact on the Raman detection sensitivity of probe molecules with relatively high concentrations. This is due to the fact that the structural properties of the hot spots primarily determine the characteristic peak intensity of the Raman signal.

According to the structural parameters corresponding to the reaction time, we simulated the distribution of the electric field intensity around the nanobowl arrays for 45 min, 2 h and 4 h, as shown in Figure 7. To illustrate the effect of the wall thickness of the nanobowl, we also simulated the electric field intensity distribution for the array with a wall thickness of 2 nm as shown in Figure 7g,h. The formation of a hot spot in the nano-bowl array occurs at the upper wall position of the bowl. The electric field intensity surrounding the solid triangle structure within the wall diminishes gradually over time. Consequently, the presence of probe molecules is more likely in the triangular location. However, the electric field intensity inside the bowl of the 4 h structure exhibits an increase, indicating a progressive enhancement of the local surface plasmon resonance effect at the inner diameter of the 75 nm bowl. Nevertheless, the overall enhancement remains constrained. Notably, the 45 min sample, characterized by the thinnest wall thickness, displays a superior enhancement in the delta region. Conversely, the structure featuring a wall thickness of 2 nm manifests an approximately tenfold increase in the electric field enhancement surrounding the triangular region compared to the 45 min sample. Figure 7a,b displays the simulation outcomes, revealing that a 14 nm wall thickness produce a “hot spot” region at the top of the wall in the depth direction, and a factor of about 10 enhances their electric field intensity. Consequently, the maximum amplitude of the enhanced electric field |E/E_0_| is about 10. Therefore, the corresponding SERS enhancement factor |E/E_0_|^4^ is approximately 10^4^.

### 3.4. SERS Enhancement Factor of AgNBs-PSS Substrate

Due to the strong conjugated absorption of Rhodamine 6G at 532 nm, this wavelength generates a strong conjugated Raman scattering. To better calculate the Raman enhancement factor, we used 4-MBA as the characterization molecule for the enhancement factor calculation (Figure 8). The EF of the AgNBs-PSS substrate was calculated using the peak intensity at 1582 cm^−1^ reflecting the C-C stretching vibration of the 4-MBA molecule. The formula for EF calculation is given by the following:(2)$$\mathrm{EF}=(I_{SERS}/N_{SERS})/(I_{bulk}/N_{bulk})$$
(3)$$N_{bulk}=\frac{\rho_{4-MBA}\times V_{Laser}}{M_{4-MBA}}\times NA$$
(4)$$N_{SERS}=\frac{S_{\mathrm{SERS}}}{A_{4-MBA}}$$
(5)$$V_{Laser}=S_{Laser}\times H_{Laser}$$

Among them, *I*_SERS_ and *I*_Bulk_ are the Raman intensities of monolayer 4-MBA on the AgNBs-PSS substrate and 4-MBA powder on a silicon substrate, respectively, and their values are 242 counts and 2320 counts, respectively. *N_SERS_* and *N_Bulk_* are the 4-MBA molecules that contribute to the Raman intensity within the laser irradiation area, respectively. For solid powder, we used the solid density *ρ*_(4-MBA)_ (1.3 g/cm^3^) of 4-MBA, the molecular mass *M_4-MBA_*(154), and the laser irradiation volume *V_Laser_* (2512 μm^3^) to obtain the *N_Bulk_* value. Among them, *V_Laser_* is calculated by the diameter of the irradiation area *S_Laser_*, which is 10 μm, and the laser focus depth *H_Laser_*, which is 32 μm measured by the adequate illumination depth collected by Raman spectroscopy. The value of *N_Bulk_* is approximately equal to 1.28 × 10^13^. Based on the SEM image, we estimate the area with a hot spot effect in the excitation light area. A single laser spot contains 5 micron cones. According to the electron microscope image, 144 nanobowls were distributed on each micron cone. The size of the nanobowl is equal to the position of the mouth of the bowl. The circumference 155π is multiplied by the wall thickness of 14 nm, and the *S_SERS_* equals 4.9 × 10^6^ nm^2^. We used the area 4.9 × 10^6^ nm^2^ of hot spots within the substrate in the laser-irradiated area and the area *A_4-MBA_* pass of a single 4-MBA molecule to evaluate *N_SERS_* values. It is known from the previous literature [38] that *A_4-MBA_* is 0.33 nm^2^. The resulting *N_SERS_* corresponds to 1.49 × 10^7^. Therefore, EF is calculated as 8.96 × 10^4^.

The spatial distribution of nanobowl hot spots is observed along the curved surface of the bowl. However, the bowl’s wall structure, as fabricated in this study, exhibits a relatively larger thickness, exceeding 10 nm, consequently leading to suboptimal enhancement effects. Additionally, the nanobowl array exhibits a compact configuration, restricting the formation of hot spot regions and resulting in a comparatively diminished substrate enhancement factor compared with other related work on nanobowls, it is shown in Table 3.

Moving forward, our research endeavors aim to fabricate nanobowls possessing wall thicknesses below 10 nm. Furthermore, we seek to design an array structure with nanogaps interposed between the bowl walls, thereby advancing the SERS enhancement effects.

## 4. Conclusions

To enhance the performance of SERS substrates, we implemented the light confinement effect by constructing a 3D-SERS substrate employing PSS as the carrier, featuring a periodic microcone structure. In this study, we report the fabrication of a 3D-SERS substrate comprising PSS as support and regular silver nano cups loaded on PSS by exploiting PS self-assembly and the surface electrochemical displacement reaction. The surface electrochemical displacement reaction is observed to gradually replace the Ag shell of the nano cups from the outside of the PS spheres due to the adsorption of Sn^2+^ by PS. The PSS exhibits a superior light-capture effect compared to flat substrates due to the periodically patterned structures. PSS has great potential to be an excellent SERS support structure. AgNBs structures and SERS performance have been optimized by controlling reaction time. Utilizing FDTD simulation, our investigation reveals that a “hot spot” predominantly resides within the boundary region of the nanobowl array structure. The performance of substrates with optimized preparation parameters was tested using 4-MBA with an EF of 8.96 × 10^4^. Moreover, we demonstrate the potential for further optimization of this hot spot configuration within the bowl wall area.

## Figures and Tables

**Figure 1 micromachines-14-01197-f001:**
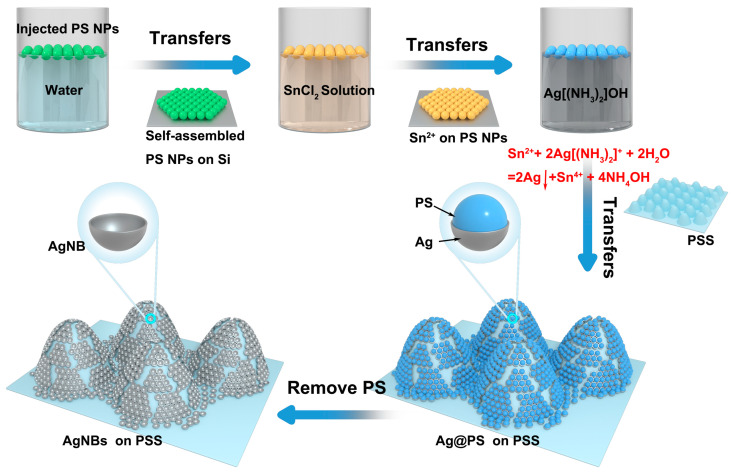
Schematic diagram of the preparation process of PSS-supported Ag nanobowl structures. Schematic diagram of the process of PS self-assembled monolayer adsorbing Sn^2+^ and electrochemical replacement to form an Ag shell fixed on the PSS; schematic diagram of the microstructural changes before and after the removal of PS by DMF dissolution.

**Figure 2 micromachines-14-01197-f002:**
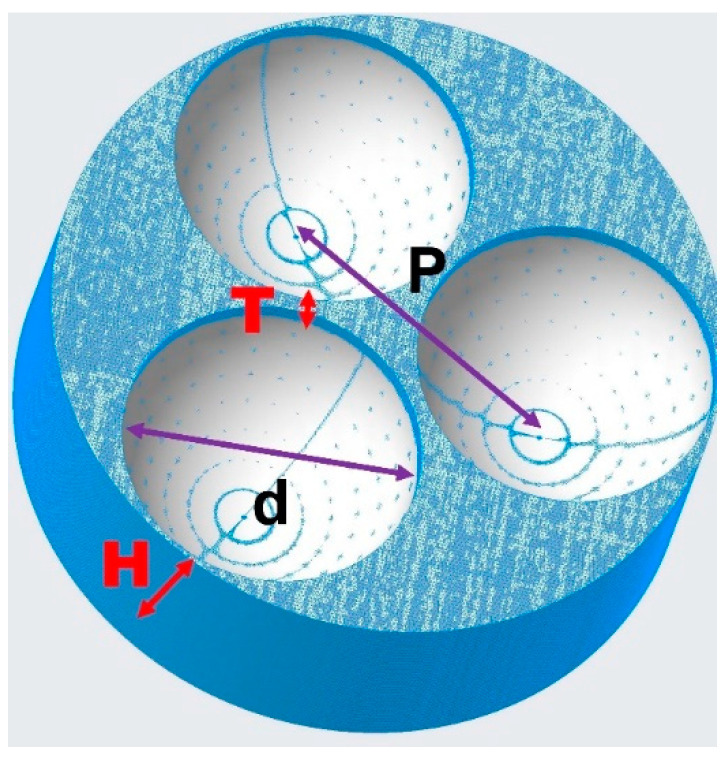
Schematic diagram of the characteristic parameters of AgNBs structure.

**Figure 3 micromachines-14-01197-f003:**
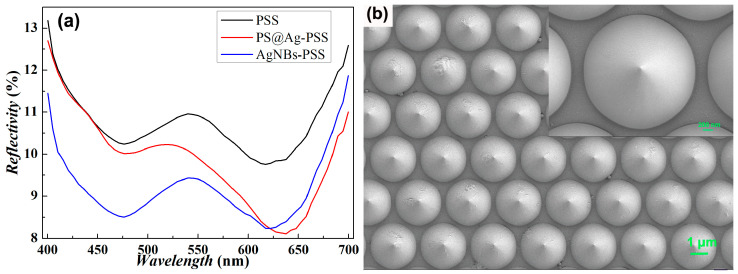
(**a**) Reflectance spectra based on SS–Al mirror substrates; (**b**) SEM structure image of PSS, with an enlarged detail of a single cone structure in the upper right corner.

**Figure 4 micromachines-14-01197-f004:**
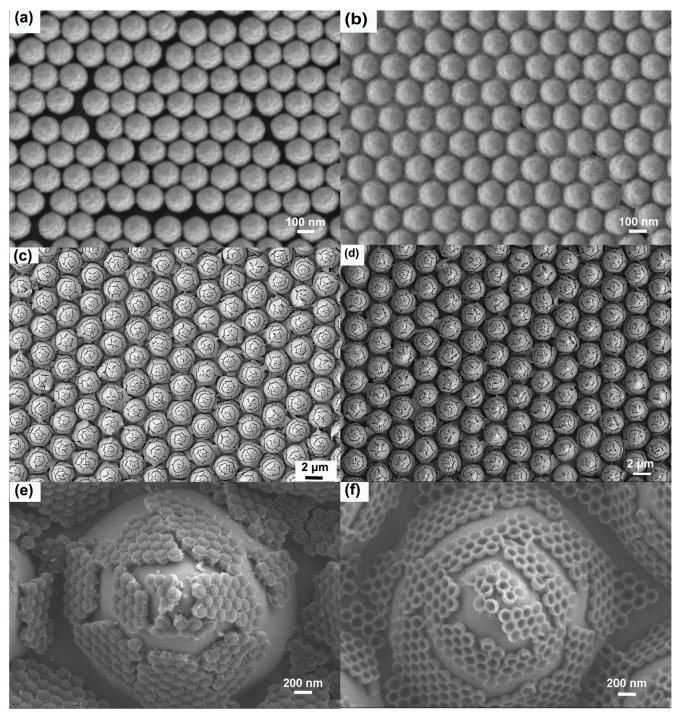
SEM images of the preparation process and corresponding PSS-supported Ag nanobowl substrate structure. (**a**) Self-assembly of PS nanospheres into a monolayer; (**b**) Adsorption of Sn^2+^; (**c**) Electrochemical displacement reaction for fixing AgNBs onto PSS substrate; (**d**) Removal of PS spheres to form AgNBs. (**e**,**f**) Detailed SEM images of nanostructures on a single PSS cone top corresponding to (**c**,**d**), respectively.

**Figure 5 micromachines-14-01197-f005:**
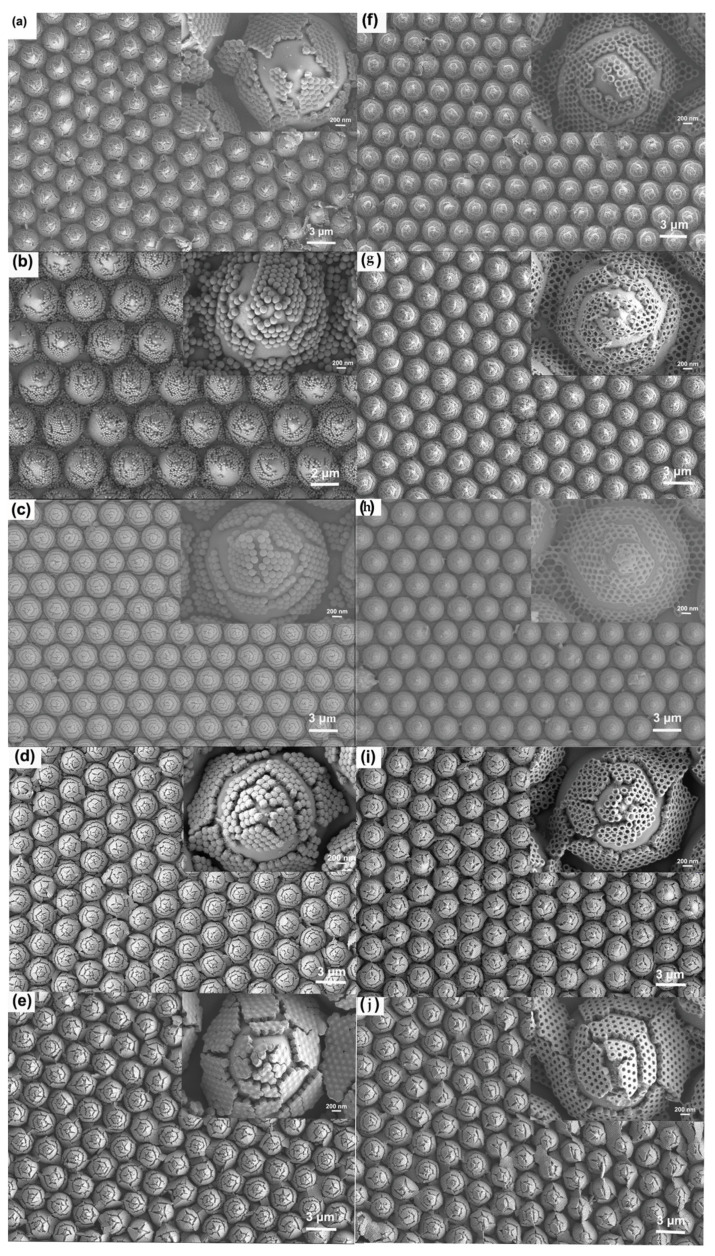
SEM images of AgNBs-PSS substrates prepared using 150 nm PS spheres with different reaction times. The inset in the upper right corner is an enlarged detail of a single cone structure. Panels (**a**–**e**) show the structures of Ag-coated PS on PSS substrates after electrodeposition for 45 min, 1 h, 2 h, 3 h, and 4 h, respectively. Panels (**f**–**j**) show the structures of AgNBs on PSS substrates after removing the PS spheres for 45 min, 1 h, 2 h, 3 h, and 4 h, respectively.

**Figure 6 micromachines-14-01197-f006:**
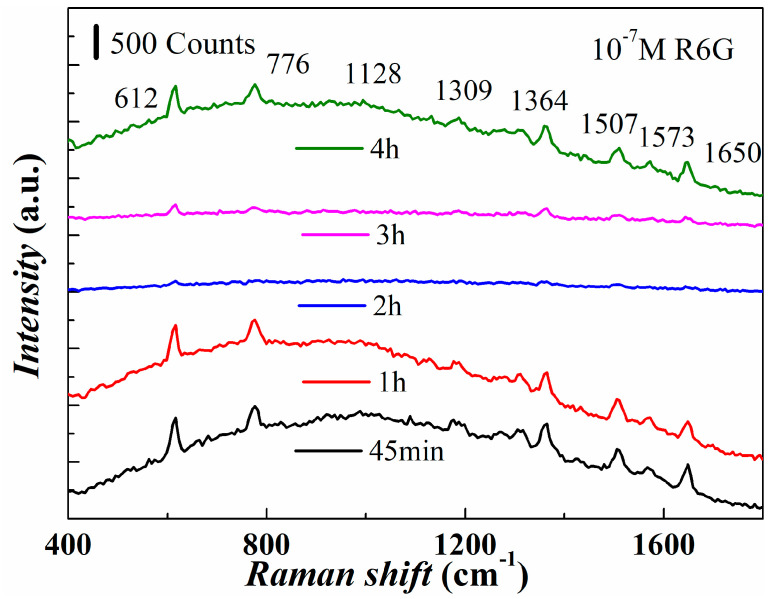
SERS plots of AgNBs-PSS substrates prepared by 150 nm PS sphere electrocoupling reaction time 45 min, 1 h, 2 h, 3 h, 4 h for testing 10^−7^ M R6G.

**Figure 7 micromachines-14-01197-f007:**
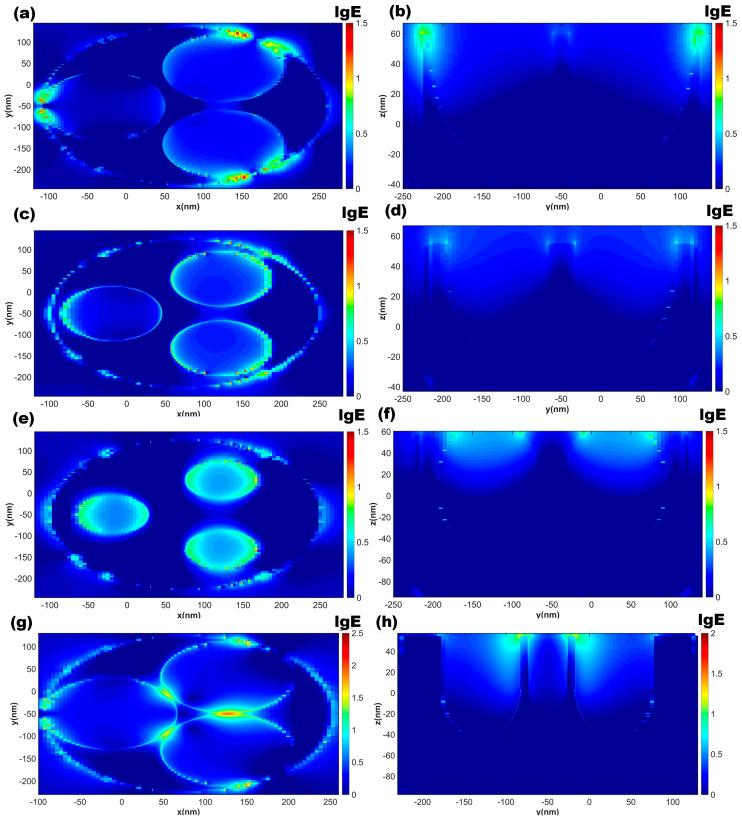
Electric field enhancement distributions of NBs cavity structures were reacted for 45 min (**a**,**b**), 2 h (**c**,**d**), and 4 h (**e**,**f**). NBs cavity structures of 2 nm wall thickness (**g**,**h**) with vertical incident light excitation in *XY* and *YZ* sections, respectively.

**Figure 8 micromachines-14-01197-f008:**
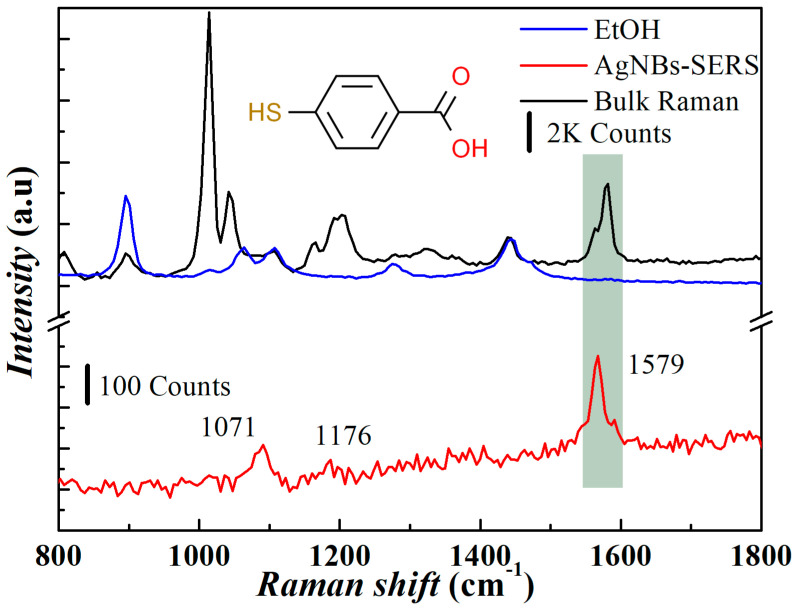
The SERS and Raman spectra of monolayer 4-MBA on the AgNBs-PSS substrate and solid 4-MBA powder on the silicon substrate.

**Table 1 micromachines-14-01197-t001:** Comparison of fabrication process characteristics.

Fabrication Process	Characteristic Parameters	Advantages	Limitations	Enhanced Capability of SERS	References
EBL/FIB *^1^	The aspect ratio of ellipses	Ordered 2D structure, good structural precision	Complex equipment, expensive preparation costs, low efficiency	5 × 10^6^	[32]
NSL *^2^	Nanopore	Ordered 2D structure, simple equipment, and low cost	Medium structural accuracy, deformable support	3 × 10^6^	[33]
NIL *^3^	Nanopore	Simple equipment and low cost	Depends on the template, deformable support	10^8^	[34]
Deposition	Nanoparticles	Simple equipment and low cost	Poor structural accuracy, disordered structure	9.7 × 10^5^	[35,36]
Self-assembly	Nanoparticles	Ordered 2D structure, simple equipment, and low cost	Medium structural accuracy	2 × 10^7^	[37]
Self-assembly on 3D PSS	AgNBs	Simple equipment and low cost	Medium structural accuracy	10^5^	This work

*^1^, Electron-beam lithography (EBL), Focused ion beam lithography (FIB); *^2^, Nanosphere lithography (NSL); *^3^, Nanoimprint lithography (NIL).

**Table 2 micromachines-14-01197-t002:** Data table of the characteristic parameters of AgNBs structures with reaction time.

Reaction Time	Inner Diameter d nm	Outer Diameter D nm	Wall Thickness T nm	Height H nm	Round Center Distance P nm
45 min	127	155	14	82	164
90 min	114	156	21	90	165
2 h	110	155	22.5	65	164
3 h	99	155	28	115	165
4 h	75	155	40	118	167

**Table 3 micromachines-14-01197-t003:** Comparison of SERS detection capabilities of nanobowl structured substrates.

Substrate Structure	Detection Limit	EF	References
Ag@Al_2_O_3_ NBs	10^−10^ M P-Tc *	6.5 × 10^6^	[39]
AgNBs/3D-Si	10^−9^ M R6G	3 × 10^7^	[31]
AgNPs on AuNBs			[40]
AuNPs in Au-AgNBs	H1N1 virus		[41]
Ag conical cavity arrays	10^−9^ M R6G	7 × 10^6^	[42]
AgNBs-PSS	10^−7^ M R6G	9 × 10^4^	This work

*, p-thiocresol (P-Tc).

## Data Availability

The authors confirm that the data supporting the findings of this study are available within the article.

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
