# Peer review of "Fabrication of Silver Nanobowl Arrays on Patterned Sapphire Substrate for Surface-Enhanced Raman Scattering"

_micromachines, 2023, doi:10.3390/mi14061197_

Round 1
Reviewer 1 Report
This manuscript describes the fabrication of a 3D surface-enhanced Raman spectroscopy (SERS) substrate considering patterned sapphire substrate (PSS) as support and regular silver nano cups adhered on PSS. However, this manuscript must be improved based on the following issues:
1.-The English redaction is hard to read. The English redaction must be enhanced in all the manuscript sections.
2.-The title of the manuscript must be modified. The terms preparation and performance study are not suitable for the content reported in the manuscript. On the other hand, the term fabrication should be included.
3.-The abstract redaction is weak. This section must include an introduction, methods and materials, main results, and conclusion. The abbreviations must be described. For instance PSS and PS.
4.-The introduction redaction is hard to read. In addition, this section should incorporate the main research problem. Also, this section must consider more critical discussions of the main limitations of other research works related to the fabrication of a 3D surface-enhanced Raman spectroscopy (SERS) substrate.
5.-The introduction should add the main advantages of the proposed fabrication process compared with others reported in the literature.
6.-This manuscript should incorporate a table with the main characteristic parameters, advantages, and limitations of the proposed fabrication process compared with others reported in the literature.
7.-The section on materials and methods is difficult to read. The description of this section should be significantly improved. The description of the fabrication process must include more detailed technical information. Subsection 2.3 is poor. This sub-section must be enhanced including more technical information on the simulation models, boundary conditions, mesh type, and analysis type.
8.- The section on results and discussion must include more information about the different parameters of the tested samples.
9.-The discussions of the main results of Figures 5 and 7 must be enhanced.
10.-The resolution of Figures 2a and 5 must be enhanced.
11.-The discussion on the results shown in Figure 6 is weak.
12.-The conclusion section must be modified including the commented changes.
13.-What is the future research work?
The English redaction is hard to read. The English redaction must be enhanced in all the manuscript sections.
Reviewer 2 Report
The paper is well presented. The figures are high quality and the writing is clear. Therefore, I strongly recommend it for publication as the current form.
Author Response
No further comments.
Reviewer 3 Report
The manuscript of the paper entitled "Preparation and Performance Study of Class-Compound Eye 2 Three-Dimensional Nanostructured SERS Substrate" presents a new method for the fabrication of a SERS substrate consisting of an array of conical microstructures coated with metal nanoparticles. The paper describes the fabrication method and the detection study with two probe molecules: R6G and 4-MBA
In my opinion, this manuscript can be published in the journal Micromachines after some minor revisions.
Regarding the introduction, as the manuscript deals with SERS, it would be interesting to explain more about the SERS effect and the definition of a hot spot (line 36, 43, 53) and a plasmon (line 44 and following) should be more detailed.
Line 40 "The use of a confocal laser beam in the SERS spectrometer reveals that 2D substrates fail to maximize the excitation light due to the 3D conical shape of the beam" : This assumption related to reference 9 could be more detailed.
Line 45 "3D SERS substrates along the z-axis can easily customize their shape, size, and periodicity of their structure, which allows to tune optical phenomena and plasmonic properties by increasing structural complexity" : This statement should be explained in detail.
Even if it is obvious, please write Polystrene to explain what PS means.
Regarding materials and methods, FDTD should be more detailed: the background is air but what was defined for the 3D system since it is not only Ag nanoparticles (the surface coverage seems to change in the SEM images)? What physical quantity is calculated?
Concerning the results and the discussion
Line 166 "The electric field ... of the local surface plasmon coupling E" should be explained from a physical point of view. Where does it come from? The coupling with what?
Figure 6 should be better described for a better understanding of the influence of the wall thickness.
Reviewer 4 Report
The work described in “Preparation and Performance Study of Class-Compound Eye Three-Dimensional Nanostructured SERS Substrate” by Yanzhao P. et al. successfully demonstrated a novel technique for fabricating three-dimensional nano-bowl arrays to enhance the SERS signal. Overall, this study is well conducted, well written, and suitable for publication in the journal of “Micromachines”. The publication is recommended after addressing the following comments and questions.
1. A title of the current article does not appear to reflect the contents of this article. In particular, the meaning of “class-compound eye” is difficult for readers who are not experts in this field. The referee recognizes that the merits of this article revealed that Ag nano-bowl structures on PSS have a potential for enhancing the SERS signal. The referee suggests reconsideration of this article's title.
2. Please add a description of the preparation of patterned sapphire substrate (PSS) to “2. Materials and Methods”.
3. Please add information to the section “The substitution-reduction reaction of Ag on the surface of PS nanoparticles” regarding when the PSS was immersed in the reaction pool.
4. In the caption of Fig. 4, the letters representing Panels are incorrect.
5. Please include a schematic model to obtain the results for the electric field enhancement distribution in Fig.6. With the current figure, it is difficult for the referee to understand what the electric field distribution in the Ag nano bowl looks like.
6. In lines 209-221, the authors discuss relations between the SERS peak intensities and the structures of the Ag nano-bowl. However, as it stands, it is not very convincing. The addition of simulations of the electric field enhancement distribution of Ag nano-bowl at reaction times of 45 min, 2h, and 4h will help readers to better understand this argument.
7. Please check the figure number on page 8. This figure should probably be Fig. 7.
8. Please add a short comment on the relationship between the SERS signal and the coverage rate of Ag nano-bowls to the surface area of PSS.
-
Round 2
Reviewer 1 Report
The authors have significantly improved their manuscript.
The English grammar is aceptable.